# Search for Non-Protein Protease Inhibitors Constituted with an Indole and Acetylene Core

**DOI:** 10.3390/molecules26133817

**Published:** 2021-06-23

**Authors:** Marco A. Almaraz-Girón, Ernesto Calderón-Jaimes, Adrián Sánchez Carrillo, Erik Díaz-Cervantes, Edith Castañón Alonso, Alejandro Islas-Jácome, Armando Domínguez-Ortiz, Sandra L. Castañón-Alonso

**Affiliations:** 1Departament de Química, Universidad Autónoma Metropolitana-Iztapalapa, San Rafael Atlixco 186, Col. Vicentina, Iztapalapa, Ciudad de México C.P. 09340, Mexico; marcofquim@gmail.com (M.A.A.-G.); aij@xanum.uam.mx (A.I.-J.); doar@xanum.uam.mx (A.D.-O.); 2Laboratory de Investigación en Inmunoquímica, Unidad de Investigación en Inmunología Proteómica, Hospital Infantil de México Federico Gómez, Calle Dr. Márquez Nº 162, Col. Doctores, Delegación Cuauhtémoc, Ciudad de México C.P. 06720, Mexico; ausbir@yahoo.com.mx (A.S.C.); dental_eca@hotmail.com (E.C.A.); 3Centro Interdisciplinario del Noreste, Departament de Alimentos, Universidad de Guanajuato, Tierra Blanca, Guanajuato C.P. 37975, Mexico

**Keywords:** synthesis, indole derivatives, inhibitor of proteases, studies *in silico*, docking studies, SARS-CoV-2-M^pro^, KLK5 protease

## Abstract

A possible inhibitor of proteases, which contains an indole core and an aromatic polar acetylene, was designed and synthesized. This indole derivative has a molecular architecture kindred to biologically relevant species and was obtained through five synthetic steps with an overall yield of 37% from the 2,2′-(phenylazanediyl)di(ethan-1-ol). The indole derivative was evaluated through docking assays using the main protease (SARS-CoV-2-M^pro^) as a molecular target, which plays a key role in the replication process of this virus. Additionally, the indole derivative was evaluated as an inhibitor of the enzyme kallikrein 5 (KLK5), which is a serine protease that can be considered as an anticancer drug target.

## 1. Introduction

Proteases are essential for replication/transmission of viruses, parasites, and bacteria that cause infectious disease in mammals. These enzymes regulate most physiological processes by controlling the activation, synthesis, and turnover of all proteins. Inhibitors of such proteases could potentially be useful in the treatment of diseases as diverse as cancer; parasitic, fungal, and viral infections (e.g., schistosomiasis, malaria, C. Albicans, human immunodeficiency virus (HIV), hepatitis, herpes); and inflammatory, immunological, respiratory, cardiovascular, and neurodegenerative disorders including Alzheimer′s disease [1]. On the other hand, proteases are classified according to their catalytic site [2]. Inhibitors can be generally classified into two large groups based on their structural dichotomy: low molecular weight peptidomimetic inhibitors and protein protease inhibitors composed of one or more peptide chains. Protease inhibitors can be further classified into five groups (serine, threonine, cysteine, aspartyl, and metalloprotease inhibitors) according to the mechanism employed at the active site of proteases they inhibit [3].

Otherwise, chymotrypsin-like serine proteases have been found in vertebrates, bacteria, and also viruses; the hepatitis C virus NS3 protease is an example of a viral protease in this family [4]. Therefore, the NS3 protease of hepatitis C virus (HCV) has been studied intensively over the past few years as a target for antiviral drugs [3,4,5]. Several research groups have explored the use of indole derivatives to be candidates for HCV protease inhibitors. Ismail [6,7] reported a series of indole derivatives, represented by **1** (Figure 1), as potent HCV NS3/4A serine protease inhibitors; molecular modeling simulation studies were used in order to predict the biological activity of the proposed compounds. Narjes et al. [8,9,10] reported a series of allosteric indole-*N*-acetamide inhibitors, typified by **2** (Figure 1), which are potent inhibitors of the HCV NS5B polymerase. Inhibition of NS5B enzyme has emerged as an attractive strategy for the development of antivirals to combat HCV because NS5B plays a crucial role in viral replication.

It is interesting to note that, lopinavir/ritonavir is a co-formulated human immunodeficiency virus (HIV)-specific protease inhibitor [11], interferon-α2b is a registered agent for chronic hepatitis C [11], umifenovir (**3**, Figure 1) has been demonstrated to inhibit in vitro infection with globally prevalent pathogenic viruses, including hepatitis C, and umifenovir has an indole core [12]. Viral proteases are well-validated drug targets that have led to various approved drugs, for example, against chronic infections with HIV or hepatitis C virus (HCV) [13], which employ aspartyl and serine proteases, respectively [1,14]. Further, several reviews have been published describing compounds containing acetylene groups and related functional groups, isolated from plants, fungi, microorganisms, and marine invertebrates, which present interesting biological activities [15,16,17]. There are acids with two groups of consecutive alkynes that act as inhibitors of HIV protease [15].

On the other hand, one of the best-characterized drug targets among coronaviruses is the main protease (M^pro^, also called 3-chymotrypsin-like cysteine protease, 3CL^pro^); inhibiting the activity of this enzyme would block the viral replication. Because no human proteases with a similar cleavage specificity are known, such inhibitors are unlikely to be toxic [18,19]. Dai et al. [20] designed indole derivatives **4** (Figure 1), which are the strongest known SARS-CoV-2 M^pro^ inhibitors, with inhibition constants in the two-digit nanomolar range [14,20]. Different drugs were tested as possible therapeutic candidates against SARS-CoV-2 (severe acute respiratory syndrome–coronavirus 2) infection [11,12], such as: remdesivir, lopinavir/ritonavir, favipiravir, umifenovir, EIDD-2801, convalescent plasma collected from donors who have survived an infectious disease, interferons, corticosteroids, vitamin C, vitamin D, zinc, the non-steroidal anti-inflammatory drugs (NSAIDs) [21], antibodies (for example, REGN-COV2) [22], among others. On the other hand, studies in silico to evaluate the potential of hundreds [23] or thousands [24,25] or millions [13] of molecules against the active site of SARS-CoV-2 have been reported [26,27,28].

Moreover, kallikrein-related peptidases (known as KLKs) constitute a single family of 15 highly conserved trypsin- or chymotrypsin-like serine proteases encoded by the largest uninterrupted cluster of protease-encoding genes (*KLK1–15)* in the human genome [29]. Aberrant KLK expression patterns have been reported mainly in hormone-dependent malignancies, such as those in breast, ovary, and prostate, and have been widely implicated as cancer biomarkers. In ovarian cancer, overexpression of KLK5 protease in tumor tissues is associated with more advanced stages and grade of the disease [30]. On the other hand, KLK5 plays a central role in the degradation of corneodesmosomes, which are the main adhesive structures in the cornified cell layer. KLK5 is also involved in the activation of other epithelial serine proteases such as kallikrein-related peptidase. Although KLK5 is important for the maintenance of skin homeostasis, its overactivation can impair the skin barrier function and contribute to the discovery of various dermatoses [31]. Kim et al. [31] reported that the indole-3-acetonitrile-4-methoxy-2-*S*-*β*-d-glucopyranoside KLK5 protease activity. Additionally, the KLK5 protein has been employed as a molecular target for drug-transport single-wall carbon nanotubes and was reported their in silico assay by Díaz-Cervantes et al. [32].

Furthermore, lipophilicity constitutes a physicochemical property of paramount importance in medicinal chemistry [33]. Lipophilicity is usually measured by the partition coefficient, Log P, or log *K_ow_* (logarithm of *n*-octanol/water partition coefficient), which plays an essential role in absorption, distribution, metabolism, and excretion (ADME) characteristics of drugs, while also affecting their pharmacodynamic and toxicological profiles, such as the extent of plasma protein binding, accumulation in tissues, and unpredictable poisonous mechanism [33,34]. Regarding the activity and log P relationship, the low log P values mean less activity of the molecule in some cases [35].

Herein, we disclose our findings on the synthesis of derivative **15** (Scheme 1), which contains an indole core and aromatic polar acetylene, as a possible potential protease inhibitor. Derivative **15** was evaluated through docking assays, in which the main protease (SARS-CoV-2-M^pro^) was used as a molecular target. This protein plays a key role in the replication process of this virus and has been reported in crystallographic data [19,36]. Despite there being other reports of recombinant virus proteases, the current work selected the non-recombinant protein [37]. Additionally, we decided to evaluate the indole-acetylene derivative **15** as an inhibitor of the KLK5 protein in order to see the scope of **15** as an inhibitor of different proteases.

## 2. Results and Discussion

### 2.1. Chemical Synthesis

Our synthesis strategy to access 2,2′-[(4-{[4-({4-[(1*E*)-3- (1*H*-indol-3-yl)-3-oxoprop-1-en-1-yl]phenoxy}acetamido)phenyl]ethynyl}phenyl)azanediyl]di(ethane-2,1-diyl)dipropionate (**15**) began with the iodination of 2,2′-(phenylazanediyl)di(ethan-1-ol) (**5**), which was treated with a mixture of KI, MeOH, and water, after which it was treated with Na_2_SO_3_ to give the iodophenyl-aminoethanol **6** with a yield of 81%. Then, a Sonogashira reaction between **6** and the alkyne **7** was performed to provide **8** in 87% yield [38]. An esterification was carried out between **8** and the propionic acid (**11**) using a propylphosphonic anhydride solution (T3P^®^) [39,40,41]; the ester **10** was obtained in 79% yield. The reduction of the nitro group of **10** was carried out using sodium hydrosulfite (Na_2_S_2_O_4_) [42,43]; the amine **11** was obtained in 82% yield. Indole derivative **14** was formed from indole **12** and acid **13** as described by Ismail et al. [6]. Finally, the condensation reaction between **11** and **14** using a propylphosphonic anhydride solution (T3P^®^) [39,40,41] allowed achievement of the formation of the amide bond, with an 83% yield. Compound **15** was obtained through the five synthetic steps with a total yield of 36.98% (Scheme 1).

Once the chemical structure of the indole-acetylene derivative **15** was confirmed, docking studies were conducted to find out the potential of molecule **15** as a coronavirus protease inhibitor COVID-19 and inhibitor of serine proteases.

### 2.2. Docking Studies with SARS-CoV-2-M^pro^

Figure 2 shows the in silico molecular couplings between the SARS-CoV-2-M^pro^ and the four selected ligands: indole-acetylene derivative **15**, co-crystallized molecule **16** (known as N3, which is a Michael acceptor inhibitor) [36], 1,5-disubstituted tetrazole-1,2,3-triazoles **17** and **18** (Figure 3) [26]. Note that, all the ligands were coupled in the right-site cavity, which contains the active site as explained below, and 1,5-disubstituted tetrazole-1,2,3-triazoles **17** and **18** were selected due to their higher selectivity to the SARS-CoV-2-M^pro^, reported by Díaz-Cervantes and co-workers [28].

The derivative for the molecular coupling is depicted in Figure 2; the main interactions are shown in Table 1. The ligands are ordered from the smallest to largest interaction with the SARS-CoV-2-M^pro^, with the **18** molecule being the best ligand for this target. This can be explained with the LE values, and it can be noticed that the ligand with better interaction presents higher Hbond and VdW interactions than the others.

Although **15** is not the best ligand in Table 1, this presents a more favorable value of LE than the N3 inhibitor **16** (so-called Co-Crystal in the present work) and exergonic energy, which means that this ligand can interact in a favorable way with the SARS-CoV-2-M^pro^. It is important to highlight in Table 1 that Co-Crystal **16**, **17**, **18**, **1a**, **1b**, **2**, **3*,*** and **4** ligands were the only molecules selected to present a comparison point of the interactions between the SARS-CoV-2-M^pro^ and the synthesized molecule **15**.

Considering that the aim of this work was to evaluate the interactions between **15** and the protease present in the pandemic virus (SARS-CoV-2-M^pro^), Figure 4 shows the position of the ligand in the active site of the selected protease. Note that the virus protein is a cysteine protease with the Cys145 (and Ser144) and that His41 is the key residue in the catalytic site, which interacts with the central site of the synthesized ligand.

Regarding the obtained pose for **15** (Figure 4), the key interactions are depicted in Figure 5. The principal residues of the SARS-CoV-2-M^pro^ that interact with **15** are the Ser1, His164, and Phe305 in the case of the hydrogen bond interactions and His41 and Glu166 for the electrostatic interactions.

Moreover, to understand the bioactive conformation obtained from molecular docking it is necessary to evaluate the ligand–target hydrophobic interactions. Figure 6 shows the position of **15** due to the hydrophobic behavior of the protein. Hydrophobic and hydrophilic surfaces are present in the active site and the molecule gets a pose dependent on these interactions, in addition to the HBond, VdW, and Elstat interactions. In this order, is clear that the central site of **15** is surrounded by hydrophobic surfaces of the viral target, at the same time that some hydrophilic moieties of this interact with the red (hydrophilic) surfaces of the target, promoting the bioactive conformation of this ligand. To better understand the ligand–target interactions and to evaluate the possible change of conformation of the target, a normal mode analysis of the target was conducted using the iMod server [44] which demonstrated that the active sites present lower modification.

Finally, to conclude the analysis of the interactions between the SARS-CoV-2-M^pro^ and **15**, the bioactive conformation of the latter was inserted into the SARS-CoV-2-M^pro^ pharmacophore model computed by Díaz-Cervantes and coworkers [26] to generate an evaluation of the fragments that need the molecule to obtain better interactions with the target. Figure 7 shows that **15** needs one Hy fragment (on the left side) and two HA fragments and one HD fragment in the upper site to obtain a better interaction with the SARS-CoV-2-M^pro^.

### 2.3. Docking Studies with KLK5 Protease

The interaction energies of **15** with another therapeutic target, the KLK5 protease, are shown in Table 2. In this case, a nanostructured drug-transport (a single-wall carbon nanotube functionalized [32] with altretamine, a so-called Tube-altre) has been used as a comparative point (Figure 8A).

Table 2 shows that the ligand efficiency of **15** is better than that obtained by the Tube-altre ligand, apparently due to the higher hydrogen bond interactions of **15** than with another molecule. Note that KLK5 is a serine protease, His57 and the Ser195 being two of the key residues in the active site, which interact with the central site of **15** (Figure 8B).

Additionally, **15** interacts with KLK5 protease in terms of hydrogen bonds through Lys60, Gly193, Asp194, Ser195, and Ser214 (see Figure 9A). At the same time, Figure 9B shows the electrostatic interaction between **15** and His57, Lys60, Asp217, and Arg224. Note that the hydrogen bond is the key interaction in the ligand–target coupling.

Moreover, the hydrophobic interactions were computed, and it is clear that **15** obtains a bioactive pose due to the hydrophobic surfaces present in the active site of the KLK5 protease, as Figure 10 depicts, which is in accordance with the hydrophobic surface study of M^Pro^. The hydrophilic surfaces of **15** are on the tips of the molecule and the hydrophobic moieties (the center of **15**) and interact with the blue surfaces of the target.

Finally, to obtain the first approach with regard to the possible biodistribution of our synthesized molecule, we conducted a logP analysis of it through the server ALOGPS 2.1 [45]. The computed logP value was compared with the reference molecules. Table 3 shows that the logP for **15** is higher than that of other molecules, which can mean a better distribution of this molecule through the cellular membranes and perhaps into the virus capsid.

## 3. Materials and Methods

### 3.1. Materials and Instrumentation

Commercial reagents were purchased from Sigma Aldrich (St. Louis, MO, USA) and were used as received. Proton nuclear magnetic resonance (^1^H NMR) spectra and carbon nuclear magnetic resonance (^13^C NMR) spectra were recorded on a Bruker Advance 400 MHz spectrometer and a Varian Unity Inova 300 MHz and 75 MHz; both are reported in parts per million and are referenced to tetramethylsilane or residual solvent. Data are reported as follows: Chemical shift, integration, multiplicity (br = broad, s = singlet, d = doublet, t = triplet, q = quartet, m = multiplet), coupling constant in Hertz (Hz). Infrared (IR) spectra were obtained using a (in total attenuated reflection: FTIR-TAR mode) Perkin Elmer Spectrum 400 and Perkin Elmer 1605 spectrometer by KBr. Data are represented in frequency of adsorption (cm^−1^). Mass spectrometry was obtained with a JEOL SMX-102 A instrument (FAB+).

### 3.2. Chemical Synthesis

#### 3.2.1. 2,2′-[(4-iodophenyl)azanediyl]di(ethan-1-ol) (**6**)

2,2′-[(4-iodophenyl)azanediyl]di(ethan-1-ol) (**6**) was prepared according to the procedure described in the reference [39].

#### 3.2.2. 2,2′-({4-[(4-nitrophenyl)ethynyl]phenyl}azanediyl)di(ethan-1-ol) (**8**)

2,2′-[(4-iodophenyl)azanediyl]di(ethan-1-ol) (**6**) (2 g, 6.52 mmol) was dissolved in TEA (20 mL) and tetrahydrofuran (THF 20 mL); then 1-ethynyl-4-nitrobenzene (**7**) (1 g, 6.79 mmol) was added to this solution and stirred. This was followed by the addition of triphenylphosphine (PPh_3_, 0.02 g, 7.61 × 10^−4^ mol), copper iodide (CuI, 0.2 g, 1.05 mmol), and chloride bis triphenylphosphine palladium (II) ([Pd(PPh_3_)_2_Cl_2_], 0.2 g, 2.85 × 10^−4^ mol). The mixture was heated at 50 °C under nitrogen for 48 h, changing its color from clear yellow to orange. Afterwards, the amine salt was separated from the reaction mixture by filtration, and the filtrate was concentrated by evaporation. The obtained solid was then washed with THF, hexane, and finally acetone in order to remove the amine salt and catalysts. The product was purified by column chromatography on silica gel (1:1 hexane/AcOEt, added 5% of MeOH) to afford the desired product in 87% yield (from **6**) as an orange powder. ^1^H NMR (400 MHz, Acetone *d*-6) δ (ppm): 8.24 (d, 2H, *J* = 8.8 Hz), 7.70 (d, 2H, *J* = 8.8 Hz), 7.39 (d, 2H, *J* = 9.2 Hz), 6.80 (d, 2H, *J* = 9.2 Hz), 3.34 (S, 2H) 3.78 (t, 4H, *J* = 5.4 Hz), 3.63 (t, 4H, *J* = 5.8 Hz), ^13^C NMR (100 MHz, Acetone *d*-6) δ (ppm): 149.2, 146.4, 133.14, 131.6, 131.2, 123.7, 111.7, 107.7, 96.9, 86.05, 59.1, 53.9. FT-IR (KBr), ν (cm^−1^): 3229, 3098, 2958, 2206, 1584, 1511, 1399, 1349, 1044, 856, 799. (FAB^+^) (*m*/*z*): 326. m.p. = 151–153 °C.

#### 3.2.3. 2,2′-({4-[(4-nitrophenyl)ethynyl]phenyl}azanediyl)di(ethane-2,1-diyl)dipropianate (**10**)

2,2′-({4-[(4-nitrophenyl)ethynyl]phenyl}azanediyl)di(ethan-1-ol) (**8**) (1 g, 3.67 mmol, 1 equiv) was dissolved in THF (5 mL); then propionic acid (**9**) (1 mL, 13.5 mmol) was added to this solution and was stirred. The resulting solution was treated with pyridine (0.5 mL, 1 equiv) and this was followed by the addition of propylphosphonic anhydride solution (T3P^®^) (2 mL, 3 equiv). The mixture was heated at 50 °C under nitrogen for 48 h; the solvent was concentrated under reduced pressure and after was neutralized with sodium carbonate solution. The product was purified by column chromatography on silica gel (7:3 hexane/AcOEt) to afford the desired product in 79% yield (from **8**) as orange oil. ^1^H NMR (300 MHz, CDCl_3_) δ (ppm): 8.17, (d, 2H, *J* = 9 Hz), 7.60 (d, 2H, *J* = 9 Hz), 7.42 (d, 2H, *J* = 9 Hz), 6.74 (d, 2H, *J* = 9 Hz), 4.27 (t, 4H, *J* = 6 Hz), 3.67 (t, 4H, *J* = 6 Hz), 2.33 (q, 4H, *J* = 9 Hz), 1.13 (t, 3H, *J* = 6 Hz). ^13^C NMR (75 MHz, CDCl_3_) δ (ppm): 174.4, 148, 146.3, 133.5, 131.7, 123.6, 111.7, 109.5, 96.5, 86.6, 61.1, 49.5, 27.5, 9.1. FT-IR (KBr), ν (cm^−1^): 3297, 2927, 2857, 2209, 1719, 1607, 1518, 1380, 1339, 1231, 861, 809.

#### 3.2.4. 2,2′-({4-[(4-aminophenyl)ethynyl]phenyl}azanediyl) di(ethane-2,1-diyl)dipropianate (**11**)

2,2′-({4-[(4-nitrophenyl)ethynyl]phenyl}azanediyl)di(ethane-2,1-diyl)dipropianate (**10**) (0.5 g, 1.629 mmol) was dissolved in MeOH (30 mL), then an excess of sodium hydrosulfite salt (NaHSO_3_, 7 g, 40 mmol) in H_2_O (40 mL) was added to this solution and stirred. The resulting solution was stirred at room temperature under nitrogen for 24 h. After, the solvent was concentrated under reduced pressure, and the residue was taken up in water (40 mL) and was extracted with AcOEt (3 × 150 mL). The ethyl acetate was dried over anhydrous MgSO4 and was concentrated under reduced pressure. The product was purified by column chromatography on silica gel (1:1 hexane/AcOEt) to afford the desired product in 82% yield (from **10**) as yellow oil. ^1^H NMR (300 MHz, CDCl_3_) δ (ppm): 7.29, (d, 2H, *J* = 9 Hz), 7.21 (d, 2H, *J* = 9 Hz), 6.63 (d, 2H, *J* = 9 Hz), 6.56 (d, 2H, *J* = 9 Hz), 4.18 (t, 4H, *J* = 6 Hz), 3.56 (t, 4H, *J* = 9 Hz), 2.25 (q, 4H, *J* = 9 Hz), 2.02 (s, 2H), 1.05 (t, 6H, *J* = 9 Hz). FT-IR (KBr), ν (cm^−1^): 3317, 3061, 2958, 2928, 2210, 1722, 1682, 1583, 1465, 1355, 1266, 1165, 735.

#### 3.2.5. {4-[(1*E*)-3-(1H-indol-3-yl)-3-oxoprop-1-en-1-yl]phenoxy} acetic acid (**14**)

{4-[(1*E*)-3-(1H-indol-3-yl)-3-oxoprop-1-en-1-yl]phenoxy} acetic acid (**14**) was prepared according to the procedure described in reference [6].

#### 3.2.6. 2,2′-[(4-{[4-({4-[(1. E)-3-(1H-indol-3-yl)-3-oxoprop-1-en-1-yl]phenoxy}acetamido)phe-nyl]ethynyl}phenyl)azanediyl]di(ethane-2,1-diyl)dipropionate (**15**)

A mixture of compound {4-[(1*E*)-3-(1*H*-indol-3-yl)-3-oxoprop-1-en-1-yl]phenoxy}- acetic acid (**14**) (0.5 g, 1.557 mmol) and 2,2′-({4-[(4-aminophenyl)ethynyl]phenyl}azanediyl) bis(ethane-2,1-diyl)dipropianate (**11**) (0.5 g, 1.225 mmol) was dissolved in THF (10 mL) and was stirred. Then, the solution was treated with pyridine (1.5 mL, 1 equiv) and was followed by the addition of propylphosphonic anhydride solution (T3P^®^) (2.5 mL, 3 equiv). The mixture was heated at 50 °C under nitrogen for 24 h and was concentrated under reduced pressure after. The product was purified by column chromatography on silica gel (1:1 hexane/AcOEt, added 5% of MeOH, followed by elution 6:4 hexane/AcOEt,) to afford the desired product in 83% yield (from **11**) as yellow oil. ^1^H NMR (300 MHz, CDCl_3_) δ (ppm): 9.87 (s, 1H), 8.83 (s, 1H), 8.33-8.30 (m, 1H), 8.18 (s, 1H), 7.87–7.82 (m, 2H), 7.79 (d, 1H, *J* = 3 Hz), 7.52–7.49 (m, 2H), 7.44–7.41 (m, 2H), 7.34–7.30 (m, 3H), 7.23–7.20 (m, 3H), 7.08–7.03 (m, 2H), 6.67–6.62 (m, 2H), 4.63 (s, 2H), 4.19 (t, 4H, *J* = 6 Hz), 3.58 (t, 4H, *J* = 9 Hz), 2.26 (q, 4H, *J* = 6 Hz), 1.05 (t, 6H, *J* = 6 Hz). ^13^C NMR (75 MHz, CDCl_3_) δ (ppm): 190.8, 174.6, 165.3, 148.2, 148.1, 147.4, 136.1, 132.5, 132.4, 131.8, 123.9, 123.7, 122.9, 122.6, 120.9, 120.1, 115.4, 111.9, 111.6, 109.6, 91.4, 87.3, 67.7, 61.4, 49.7, 29.9, 27.7, 9.7. FT-IR (KBr), ν (cm^−1^): 3350, 3041, 2977, 2940, 2202, 2124, 1736, 1680, 1602, 1520, 1439, 1383, 1038, 1155, 827, 812.

### 3.3. Computational Assays

To model and optimize the studied ligands, the Gaussian package (G09) [46] at the PM6 [47] level was employed. The crystallographic data of the targets, SARS-CoV-2-M^pro^ and KLK5, were obtained via the protein data bank resource, with the PDB codes 6LU7 and 2PSY, respectively. At the same time, the proteins were corrected, and a pH adjustment using the Chimera software was performed [48].

Once the ligands and targets were modeled, optimized, and corrected, the in silico molecular couplings were performed with the Molegro Virtual Docker package (MVD) [49] using the MolDock [50] scoring function. To complete the theoretical results, a pharmacophore model [26] for the SARS-CoV-2-M^pro^ was employed to overlap the synthesized ligand using the ZINCPharmer server [51].

## 4. Conclusions

The computational study showed that the indole and acetylene fragments in our molecule is a molecular motif that could be explored to further the design of plausible pharmacophores of varied interest for new pharmaceuticals that could be used to combat existing diseases or future pandemics that are a priority for humankind. One of the strategies used in this exploration was the search for new and better protease inhibitors. We carried out the synthesis of the hybrid compound **15** in our search for molecules that may have a synergy with an indole and acetylene core. Through in silico assays, it has been demonstrated that **15** is a molecule with a plausible inhibition activity against the SARS-CoV-2-M^pro^, presenting a higher interaction than the N3 inhibitor reported in other studies. At the same time, **15** can be considered as an anticancer agent, due to the favorable interactions with the *KLK5*, a protein present in ovarian cancer. Finally, **15** can be proposed as a protease inhibitor, considering the present computational assays.

## Data Availability

The data presented in this study are available in this article.

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
