# Peer review of "Search for Non-Protein Protease Inhibitors Constituted with an Indole and Acetylene Core"

_molecules, 2021, doi:10.3390/molecules26133817_

Round 1

Reviewer 1 Report

Despite the indole core the inhibitor designed by the authors have been derivatized with lateral groups that confer a specific specificity for the Mpro or 3CLpro of SARS-CoV-2, they did not fully specify that this protease is cysteine protease but with chymotrypsin like activity. A non conventional catalytic cysteine residue in 3CLpro makes it unique as compared to other chymotrypsin like enzymes and other Ser (or Cys) hydrolases. In addition, the 3CLpro consist of a catalytic Cys145-His41 dyad instead of a canonical Ser(Cys)-His-Asp(Glu) triad of chymotrypsin like proteases. Hence the hydrophobic needs at the P1 site is well interpreted by the imidazole and the inhibitory leaving group is not well described by the docking mechanism leaving the reader to interpret the docking image of the 3D structure of the inhibitor 15  and the SARS-CoV-2 Mpro.

In the future, validation of the docking data need to be performed with a recombinant Mpro (i.e. MBP_MPro) according to other authors: https://www.nature.com/articles/s42003-020-01577-x 

Author Response

Reviewer 1 wrote:

Despite the indole core the inhibitor designed by the authors have been derivatized with lateral groups that confer a specific specificity for the Mpro or 3CLpro of SARS-CoV-2, they did not fully specify that this protease is cysteine protease but with chymotrypsin like activity. A non conventional catalytic cysteine residue in 3CLpro makes it unique as compared to other chymotrypsin like enzymes and other Ser (or Cys) hydrolases. In addition, the 3CLpro consist of a catalytic Cys145-His41 dyad instead of a canonical Ser(Cys)-His-Asp(Glu) triad of chymotrypsin like proteases.

Our answer is:

We acknowledge for your comments and regarding this, we allow ourselves to reply as follows:

  • The chymotrypsin-like cysteine behavior of the Mpro is specified in lines 70-72.
  • To complement the specification of this enzyme behavior, some lines (starting in line 171-173) to explain better this point have been added.

Reviewer 1 wrote:

Hence the hydrophobic needs at the P1 site is well interpreted by the imidazole and the inhibitory leaving group is not well described by the docking mechanism leaving the reader to interpret the docking image of the 3D structure of the inhibitor 15 and the SARS-CoV-2 Mpro.

Our answer is:

Thank you very much for this warning. In consequence, the interpretation and explanation of the Figure 5 has been extended (lines 179 to 181).

Reviewer 1 wrote:

In the future, validation of the docking data need to be performed with a recombinant Mpro (i.e. MBP_MPro) according to other authors: https://www.nature.com/articles/s42003-020-01577-x

Our answer is:

We agree. Thus, the needing of recombinant targets has been highlighted in lines 111-114.

Reviewer 2 Report

The authors of the manuscript titled "Search for Non-Protein Protease Inhibitors Constituted with an Indole and Acetylene Core" have investigated the potential usefulness of a new compound bearing an indole core and aromatic polar acetylene as infibitors of SARS-CoV-2 Mpro protease and kallikrein-related peptidase 5 (KLK5).

Unfortunately, the quality of this work is rather low and the manuscript is written with little care (e.g. klk5 and KLK5 are in principle not interchangeable. And importantly, protein names are not italicized, in contrast to names of genes).

Major weaknesses include

  1. Lack of Molecular Dynamics Simulations
  2. Lack of at least in vitro SARS-CoV-2 Mpro inhibition activity
  3. No rationale to include KLK5 together with SARS-CoV-2 Mpro
  4. Table 1: no rationale to include N3 inhibitor, 1e, and P8 compounds for comparison (why authors did not compare compound 15 with compounds presented in Fig 1)
  5. Lack of SARS-CoV-2 Mpro detailed descriprion
  6. Many grammatical errors

Author Response

Reviewer 2 wrote:

Unfortunately, the quality of this work is rather low and the manuscript is written with little care (e.g. klk5 and KLK5 are in principle not interchangeable. And importantly, protein names are not italicized, in contrast to names of genes).

Our answer is:

Thank you very much for your comments. Please, apologies for these typos/mistakes. For the new version, we took all precautions to avoid errors. We have corrected this error in lines 84-86, and throughout the manuscript.

Reviewer 2 wrote:

Lack of Molecular Dynamics Simulations

Our answer is:

Thank you very much for your comment. However, we respectfully consider that. Considering this point as the first approach docking study and still more complementary studies could be done and the low-costs associated to this computational method, the possible change in the conformation of the targets has been performed by an analysis of the normal modes of the protein, which demonstrate that the active sites do not present significant variation. These ideas were explained in lines 157-163.

Reviewer 2 wrote:

Lack of at least in vitro SARS-CoV-2 Mpro inhibition activity

Our answer is:

We absolutely agree with this observation. Unfortunately, the experimental part (synthesis) was performed early in 2020, just before COVID-19 pandemic restrictions started. Indeed, our institutions remain closed up to this date due to the same reasons, making not possible to perform further experimental studies. Additionally, to perform in vitro tests, a type-4 laboratory would be needed. We do not have access to one.

Reviewer 2 wrote:

No rationale to include KLK5 together with SARS-CoV-2 Mpro

Our answer is:

Thank you very much for your comment. The reason to evaluate KLK5 and SARS-CoV-2-MPro is explained now in lines 114-116.

Reviewer 2 wrote:

Table 1: no rationale to include N3 inhibitor, 1e, and P8 compounds for comparison (why authors did not compare compound 15 with compounds presented in Fig 1)

Our answer is:

Thank you very much for this comment. The rationale for including the compounds N3 (Now 16), 1e (now 17) and P8 (now 18) can be found in the lines 140-1346. In addition to this, we have included the results from calculations of such compounds in the Figure 1, and even in the Table 1. The compounds of Figure 1 were modeled and evaluated as reference.

Reviewer 2 wrote:

Lack of SARS-CoV-2 Mpro detailed descriprion

Our answer is:

A more detailed description of that protein is now included in lines 70-74.

Reviewer 2 wrote:

Many grammatical errors.

Our answer is:

Please, accept our apologies for these mistakes. The grammar and style of the new version of our manuscript was revised by a colleague who is native speaker of English language.

Sincerely,

Dra. Sandra Luz Castañón Alonso

Departamento de Química, CBI.

Universidad Autónoma Metropolitana-Unidad Iztapalapa

Ciudad de México, México, 09340.

Phone: +52-55-5804-4600

Reviewer 3 Report

Manuscript ID: molecules-1169736

          The article with title “Search for Non-Protein Protease Inhibitors Constituted with an Indole and Acetylene Core” by Ernesto Calderón-Jaimes, Erik Díaz-Cervantes, and Sandra Luz Castañón-Alonso et al discusses the design and synthesis of possible proteases inhibitor obtained from indole core and aromatic polar acetylene. Indole derivative was evaluated through docking assays, main protease (SARS-CoV-2-Mpro) was used as a molecular target, which plays a key role in the replication process of this virus. Also, Indole derivative was evaluated as inhibitor of a kallikrein family protein, which is a subgroup of serine proteases, and can be considered as an anticancer drug target of the klk5.

General comments-

          Overall, I think the introduction is confusing as such. As per the title, the main topic seems to be based on the synthesis of indole linked with acetylene core for search of non-Protein Protease Inhibitors. While the beginning of introduction comes directly to the corona virus main protease (Mpro, also called 3-chymotrypsin-like cysteine protease, 3CLpro).

          Followed by the discussion in the introduction which leads the reader towards the importance of indole nucleus for drug development and Line 55-60 and 64-69; discussion on the synthesis of compound 3 and compound 4 seems out of place. Any specific reasoning for including this information? Similarly why no discussion was provided for compounds 1 and 2?

          Again there is a break in the introduction to the objectives of the study followed by the additional reason for using klk5 in the study (lines 84-89).

          I strongly feel the structuring of entire introduction should begin with reason for indole nucleus and acetylene core (using the revised information from lines 43-73 as per my above observations and lines 69-73). Followed by the lines 28-43 (corona virus main protease) and jointly introduce the klk5 serine protease. A strong reasoning for why the corona virus main protease and klk5 was studied should also be included in the introduction. Finally state the objectives of present study from lines 80-85 in a revised form.

          The Chemical synthesis part need to be rewritten. What I find is that the molecular weight of target molecule 15 is 712, while it has been reported as 322. Please check if the Molecular formula C43H42N3O7 corresponds to the reported molecular weight. I suggest to cross check the NMR and mass data in the chemical sections. Some places 1H NMR and other places 13C RMN has been used. These notation should be uniform/identical (1H NMR and 13C NMR). It was surprising to note that the NMR data was recorded on Varian Unity Inova having 300-MHz and 75-MHz frequency but all 13C NMR data is reported as 100 MHz? Please verify all NMR data for complete correctness before re-submission. In the NMR data for compound 15, I couldn’t find the doublet for the trans olefinic protons? Could the authors identify/assign all the protons and carbons with numbering for clarity?

Product 10 and 11 have no mass data characterization? This is very essential for new products being reported or reference to previous reports should be indicated.

At certain places mass data is reported as FAB or IE, the type of ionization used in the JEOL SMX-102 A instrument was not mentioned in the material and methods section.

Ligand structures that is, co-crystallized molecule (N3 inhibitor), 1e, and P8 compounds should be indicated as one of the figures. Why these ligands were structurally selected for the docking study should be clearly specified. A lipophilicity study would be warranted to compare the lipophilic nature of studied compound with the other molecules used for docking.

Also the molecular weights/ lipophilicity parameters for indoles used in the introduction could be presented.

No in-vitro studies were executed/planned in the present paper. Why?

          The entire manuscript needs a proper grammatical check from native English speaker or an English language editing service.

                   Please check the references section for correctness and include the same DOI format for all references. (https://doi.org/xxxxxxxxxx)

Round 2

Reviewer 2 Report

The manuscript was improved and might be published in its present form.